# Non-Coding RNAs as Critical Modulators of Cholesterol Metabolism in Cancer

**DOI:** 10.3390/biomedicines13071631

**Published:** 2025-07-03

**Authors:** Chunyu Zhang, Zhiwei Miao, Yan Xu, Tongguo Shi

**Affiliations:** 1Department of Gastroenterology, Zhangjiagang TCM Hospital Affiliated to Nanjing University of Chinese Medicine, 77 South Changan Road, Zhangjiagang 215600, China; 20234132056@stu.suda.edu.cn (C.Z.); zjgzy011@njucm.edu.cn (Z.M.); xuyan_1975@126.com (Y.X.); 2Jiangsu Institute of Clinical Immunology, The First Affiliated Hospital of Soochow University, 178 East Ganjiang Road, Suzhou 215000, China

**Keywords:** cholesterol metabolism, non-coding RNAs, cancer

## Abstract

Cholesterol metabolism reprogramming helps tumor cells meet their high energy and biosynthetic needs. Many studies link high cholesterol levels to a higher risk of cancers, such as breast, prostate, and colorectal cancer. Dysregulated cholesterol metabolism contributes to cancer development and progression. Various non-coding RNAs (ncRNAs), such as miRNAs, lncRNAs, circRNAs, piRNAs, and tRNAs, are key players in this process. However, systematic reviews of ncRNAs’ functions in cholesterol metabolism and their impact on tumor progression are limited. This review aims to address this gap by summarizing the current understanding of how ncRNAs govern cholesterol metabolism in cancer. We provide a comprehensive overview of cholesterol metabolism reprogramming in tumor progression through its influence on growth, metastasis, drug resistance, and immune evasion. Moreover, we summarize recent advances in understanding how ncRNAs regulate cholesterol metabolism in cancer, highlighting potential therapeutic targets for cancer treatment.

## 1. Introduction

The complex and multifaceted characteristics of cancer pose significant challenges for both researchers and healthcare professionals [1]. Among these challenges, the alteration of cancer-cell-intrinsic metabolism is of particular importance [2,3]. Notably, cholesterol metabolism reprogramming serves as a crucial adaptive mechanism in tumor cells, enabling them to satisfy the elevated energy and biosynthetic requirements associated with uncontrolled growth [4,5]. Consequently, the significant roles of cholesterol metabolism in cancer have driven the emergence of ways of screening new molecules and strategies targeting cholesterol metabolism as a focal point in cancer research, leading to substantial progress in recent years [6].

Cholesterol (C_27_H_46_O), first separated from human gallstones over two centuries ago, remains a subject of fascination for scientists and clinicians [7,8]. As with other sterols, cholesterol is hydrophobic and synthesized within mammalian cells. It primarily resides in cell membranes, where it interacts with surrounding lipids to regulate bilayer rigidity, fluidity, and permeability. Additionally, cholesterol binds to various transmembrane proteins, helping to maintain or alter their conformation. Beyond its role in membrane structure and function, cholesterol is converted into diverse oxysterols via enzymatic and nonenzymatic pathways; some are further metabolized into bile acids. The oxidative cleavage of cholesterol side chains produces pregnenolone, a precursor to all other steroid hormones [9,10].

The metabolism of cellular cholesterol is a highly intricate and delicately balanced system that encompasses multiple stages, namely, endogenous synthesis, exogenous uptake, conversion and storage, and excretion [9,11]. Within the cellular milieu, cholesterol synthesis predominantly occurs in the endoplasmic reticulum and necessitates the catalytic activity of a variety of enzymes, with HMG-CoA reductase functioning as the critical rate-limiting enzyme [9,11]. Exogenous cholesterol uptake is principally mediated by low-density lipoprotein (LDL) receptor (LDLR)-dependent endocytosis [12]. Cholesterol sulfate can impede cholesterol uptake by interfering with LDL receptor internalization [9,12]. Intracellular cholesterol conversion and storage primarily occur through the process of esterification, resulting in the formation of cholesterol esters that are stored in lipid droplets. Cholesterol excretion is facilitated via efflux pathways, which encompass both passive diffusion and active transport mechanisms [9]. Active transport is predominantly mediated by ATP-binding cassette (ABC) transporters, such as ABCA1 and ABCG1, which facilitate the transfer of cholesterol from cells to high-density lipoprotein (HDL), thereby promoting reverse cholesterol transport [13]. Reverse cholesterol transport represents a critical pathway for cholesterol elimination and involves the coordinated actions of the liver and intestines. Recent studies have underscored the significant role of the intestine in non-biliary cholesterol secretion, a process referred to as transintestinal cholesterol excretion [14]. Hence, a more profound comprehension of the regulatory mechanisms for cholesterol metabolism is crucial for the development of novel therapeutic strategies aimed at modulating cholesterol levels and mitigating the risks associated with cancers.

Non-coding RNA (ncRNA) encompasses transcriptionally active RNA molecules that do not undergo ribosomal translation into proteins [15,16]. These epigenetic regulators are pivotal in orchestrating transcriptomic plasticity through multidimensional regulatory axes, including cis-/trans-acting gene expression modulation, chromatin topology remodeling, alternative RNA splicing, and post-transcriptional stabilization [17]. Based on phylogenetic conservation, secondary structure motifs, subcellular compartmentalization (e.g., nuclear-enriched vs. cytoplasmic-resident), and mechanistic divergence, ncRNAs are systematically classified into evolutionarily conserved categories, such as endogenous competing microRNA (miRNA), PIWI-interacting RNA (piRNA), long non-coding RNA (lncRNA), and covalently closed circular RNA (circRNA) [17]. Emerging evidence underscores the metabolic reprogramming governed by ncRNA-driven regulatory networks across malignancies. For instance, lncRNA ENO1-IT1 functioned as a molecular guide for lysine acetyltransferase 7 (KAT7) histone acetyltransferase, dictating the histone modification pattern on its target genes, such as enolase 1 (*ENO1*), and thereby modulating glycolysis in colorectal cancer (CRC) [18]. Additionally, Wu et al. demonstrated that circRIC8B governs lipid metabolic derangements and progression in chronic lymphocytic leukemia (CLL) via the miR-199b-5p/lipoprotein lipase (LPL) axis [19]. Recently, a growing body of research has shed light on the significant role of ncRNAs in regulating cholesterol metabolism across multiple cancers [20,21]. However, there is a paucity of literature providing a systematic review of ncRNAs’ functions in cholesterol metabolism and their impact on tumor progression.

This review aims to fill this gap by summarizing the current understanding of how ncRNAs govern cholesterol metabolism in cancer. The insights gained from this review are anticipated to facilitate the identification of potential novel therapeutic strategies for cancer treatment.

## 2. Cholesterol Metabolism Is Associated with Cancer Progression

Cholesterol metabolism reprogramming is a pivotal adaptive mechanism in tumor cells, enabling them to meet the heightened energy and biosynthetic demands associated with unchecked growth [6]. Numerous epidemiological studies have unveiled a positive correlation between elevated cholesterol levels and an increased risk of various cancers, such as breast, prostate, and CRC [22,23,24]. Diets high in fat and cholesterol are also considered potential risk factors for certain malignancies [25,26]. Importantly, cholesterol metabolism plays a crucial role in tumor progression, influencing various aspects of cancer biology, including tumor growth, metastasis, drug resistance, and immune evasion (Figure 1).

### 2.1. Cholesterol Metabolism Is Involved in the Growth of Tumor Cells

In the early phase of tumorigenesis, cancer cells exhibit a heightened reliance on cholesterol, which is indispensable for the synthesis of cell membranes to sustain their rapid proliferation [27,28]. Cholesterol and its derivatives also function as key modulators of critical signaling pathways, such as Hedgehog, Wnt/β-catenin, and STAT3, thereby driving tumor initiation and progression [29,30,31]. Inhibiting cholesterol synthesis or uptake has been shown experimentally to be able to effectively curb tumor cell proliferation and growth, whereas exogenous cholesterol supplementation can exacerbate the malignant phenotype of cancer cells. For instance, a study demonstrated that knocking out the proprotein convertase subtilisin/kexin type 9 (*PCSK9*) gene can inhibit the development of lung cancer [32]. Additionally, research has found that in bladder cancer, cytochrome P450 family 27 subfamily A member 1 (CYP27A1) inhibits the proliferation of bladder cancer cells by regulating cholesterol homeostasis [33]. Moreover, lymphatic endothelial-like cells are present in glioblastomas and promote the growth of CCR7-positive glioblastoma stem cells through CCL21-driven cholesterol metabolism [34]. Hence, targeting cholesterol metabolism may be a potential strategy for inhibiting tumor cell proliferation.

### 2.2. Cholesterol Metabolism Regulates the Migration and Invasion of Tumor Cells

Cholesterol metabolism plays a critical role in regulating the migration and invasion of tumor cells [35,36]. Accumulating evidence has shed light on this intricate relationship from multiple dimensions. Cholesterol enhances the risk of the epithelial–mesenchymal transition (EMT) in cancer. Fu et al. observed that sterol O-acyltransferase 1 (SOAT1) is positively related to the poor prognosis of hepatocellular carcinoma (HCC) and EMT markers, and it promotes cell migration and invasion in vitro. This process is mediated by increased cholesterol levels in the plasma membrane and the accumulation of cholesterol esters [37]. Cholesterol also modulates cell metastasis via key signaling pathways. In clear cell renal cell carcinoma (ccRCC), cholesterol treatment significantly enhances cell migration and invasion by regulating the KLF5/miR-27a/FBXW7 pathway [38]. Additionally, certain oxysterols emerge as key players in this process. For instance, 27-hydroxycholesterol (27HC), synthesized through the oxidation of cholesterol, has been proven to boost the invasive and metastatic capacity of breast cancer cells. It exerts this effect by increasing the expression of matrix metalloproteinase 9 (MMP9) and inducing EMT via the activation of the STAT3 signaling pathway, as demonstrated by Shen et al. [39]. In summary, cholesterol metabolism is a critical regulator of the migration and invasion of tumor cells in various cancers.

### 2.3. Cholesterol Metabolism Is Associated with the Drug Resistance of Tumor Cells

In terms of drug resistance, tumor cells adapt to therapeutic pressures by modulating cholesterol metabolism. Multiple lines of evidence have uncovered the intricate relationship between cholesterol metabolism and drug resistance in cancers [40,41]. In breast cancer, using a cholesterol-depleting agent (acetyl plumbagin) alongside tamoxifen increases apoptosis and reduces intracellular cholesterol in breast cancer cells. This disrupts the PI3K/Akt/PKB and Akt/mTORC1 pathways in breast cancer cells, thereby inhibiting cell proliferation and overcoming resistance [42]. In non-small cell lung cancer (NSCLC), the long-term use of EGFR-TKIs leads to drug resistance, characterized by cholesterol accumulation in lipid rafts. This accumulation boosts EGFR/Src interaction, reactivates EGFR/Src/ERK signaling, and triggers SP1 nuclear translocation and ERRα re-expression. Combining gefitinib with cholesterol-lowering lovastatin synergistically inhibits gefitinib-resistant cells, offering a promising therapy for such resistance [43]. In HCC, activation of the Sterol regulatory element-binding protein 2 (SREBP2)–3-Hydroxy-3-methylglutaryl-CoA reductase (HMGCR) axis can enhance the cholesterol biosynthesis pathway, leading to drug resistance. Research indicates that the caspase-3-induced activation of SREBP2 drives drug resistance by promoting cholesterol biosynthesis in HCC [44]. Thus, understanding these mechanisms of cholesterol metabolism is vital for developing new therapies to overcome chemoresistance in cancers.

### 2.4. Cholesterol Metabolism Modulates the Immune Response in Cancer

Cholesterol metabolism significantly impacts the tumor microenvironment and the immune response. Alterations in tumor cell cholesterol metabolism can disrupt local cholesterol homeostasis, increasing extracellular cholesterol uptake and generating metabolites that deplete cholesterol from immune cells [6,45,46]. This compromises immune cell function, enabling tumor cells to evade immune surveillance [6,45,46]. Recent studies show that in tumor-infiltrating CD8+ T cells, cholesterol levels correlate positively with immune checkpoint markers (e.g., PD-1, 2B4, TIM-3, LAG-3). Cholesterol upregulates these markers by enhancing ER stress and activating XBP1, which regulates *PD-1* and *2B4* transcription. Inhibiting XBP1 or reducing cholesterol in CD8+ T cells can restore their anticancer activity [47]. Oxysterols in the tumor microenvironment also affect T cell function by promoting reciprocal regulation between the LXR and SREBP2 pathways, leading to cholesterol depletion, metabolic abnormalities, T cell exhaustion, and dysfunction. However, increasing cholesterol in CAR-T cells by blocking LXR can enhance their anticancer function [48]. Moreover, a deficiency in RIPK3 results in cholesterol abrogation within myeloid-derived suppressor cells (MDSCs), thereby facilitating the activation of tumor-infiltrating MDSCs. This highlights the therapeutic potential of targeting cholesterol synthesis as a strategy to overcome tumor immune evasion [49]. In nonalcoholic steatohepatitis (NASH)-derived HCC, androgen receptor (AR)-driven oncogene cell-cycle-related kinase (CCRK) induces the STAT3–AR axis, activates the mTORC1 cascade, and enhances MDSC recruitment and tumorigenicity [50]. Therefore, targeting cholesterol metabolism offers a promising approach to counteract tumor immune evasion and improve immune-mediated cancer therapies.

Overall, cholesterol metabolism profoundly impacts tumor progression by influencing growth, metastasis, drug resistance, and immune evasion. Understanding these mechanisms is crucial for developing novel therapeutic strategies targeting cholesterol metabolism to combat tumors.

## 3. ncRNAs Regulate Cholesterol Metabolism in Cancers

ncRNAs, including miRNA, lncRNA, and circRNA, have been extensively documented as key regulators of gene expression, cellular physiology, and tumor development [20,51,52]. Notably, these ncRNAs also play a pivotal role in modulating cholesterol metabolism within cancer cells [53].

### 3.1. MiRNAs Are Involved in Modulating Cholesterol Metabolism in Cancers

MiRNAs are endogenous ncRNA molecules approximately 20–22 nucleotides in length. They guide the RNA-induced silencing complex to target mRNAs through partial complementarity to the 3′ untranslated region (3′ UTR), thereby promoting mRNA degradation [51,52]. MiRNAs can also inhibit translation by blocking ribosome recruitment and movement on target mRNAs, likely by interfering with the eIF4F complex, thus reducing protein synthesis. As crucial post-transcriptional regulators, miRNAs play key roles in gene expression, cell physiology, and tumor development [51,52]. Notably, there is a strong association between miRNAs and cholesterol metabolism in cancers, as summarized in Table 1 and Figure 2.

#### 3.1.1. MiRNAs Modulate Cholesterol Biosynthesis in Cancers

Accumulating evidence demonstrates that key enzymes in cholesterol biosynthesis—including HMGCR, 3-hydroxy-3-methylglutaryl-CoA synthase 1 (HMGCS1), Farnesyl-diphosphate farnesyltransferase 1 (FDFT1), and Squalene epoxidase (SQLE)—exhibit not only tumor-specific overexpression but also sophisticated regulation by miRNAs across malignancies [54]. Notably, HMGCR, the primary rate-limiting enzyme in cholesterol homeostasis, has been mechanistically linked to metastatic progression. Specifically, breast-cancer-derived extracellular vesicles transport miR-9-5p to simultaneously target insulin-induced gene 1 (INSIG1)/INSIG2 (endogenous HMGCR suppressors) and activating transcription factor 3 (ATF3), thereby upregulating HMGCR-mediated cholesterol synthesis in coordination and enhancing 25-hydroxycholesterol (25-HC) conversion. This dual regulatory mechanism critically promotes hepatic and pulmonary metastases in breast cancer models [55].

Intriguingly, the mevalonate pathway’s downstream effector FDFT1 has recently emerged as a multifunctional oncogenic driver implicated in both tumor progression and therapy resistance [56]. Azhar et al. revealed that miRNA-146b-5p-mediated FDFT1 suppression preferentially redirects metabolic flux through the non-sterol mevalonate branch, establishing a novel lipid metabolism–chemoresistance axis in bladder cancer through cisplatin tolerance induction [57]. Furthermore, SQLE overexpression demonstrates prognostic significance in advanced prostate cancer, where miR-205-mediated SQLE downregulation effectively suppresses de novo cholesterol synthesis, cellular proliferation, and resistance to next-generation androgen receptor inhibitors [58]. Of clinical significance, You et al. identified hsa-miR-363-3p as a master regulator of SQLE in pancreatic adenocarcinoma (PADD), with its tumor-suppressive effects extending beyond metabolic reprogramming to modulate immune checkpoint expression (PD-1/CTLA-4) and reshape the tumor microenvironment [59].

Notably, the cholesterol synthesis axis demonstrates therapeutic vulnerability through miRNA-mediated interventions. 24-dehydrocholesterol reductase (DHCR24), another critical biosynthetic enzyme, is effectively targeted by hucMSC-sEV-delivered miR-370-3p, which suppresses cervical cancer progression via cholesterol homeostasis disruption [60].

Collectively, these findings demonstrate that miRNA networks are pivotal in regulating cholesterol biosynthesis, with their influence extending from direct enzymatic control to modulation of the tumor microenvironment. As our understanding of the molecular mechanisms underlying the miRNA-mediated regulation of key enzymes, such as HMGCR, HMGCS1, FDFT1, and SQLE, continues to grow, it becomes evident that future research should focus on comprehensively mapping the miRNA regulatory networks targeting cholesterol biosynthetic enzymes across diverse cancer types. By elucidating the intricate interplay between miRNAs, cholesterol metabolism, and tumor biology, we can uncover novel therapeutic opportunities and ultimately improve patient outcomes.

#### 3.1.2. MiRNAs Modulate Uptake and Efflux of Cholesterol in Cancers

Emerging evidence highlights the dual regulatory axes governing cholesterol homeostasis—exogenous uptake and intracellular efflux—as critical determinants of tumor progression [10,61]. Specifically, exogenous cholesterol uptake involves the LDLR-mediated internalization of LDL particles to fulfill cellular metabolic demands [62]. Intriguingly, Wang et al. demonstrated that IDH-mutant gliomas exhibit paradoxical cholesterol dynamics; these tumors secrete cholesterol-rich particles to generate lipid-laden, pro-inflammatory, glioma-associated microglia/macrophages (GAMs) without altering GAMs’ intrinsic biosynthesis pathways. This cholesterol hypersecretion coincides with reduced tumoral cholesterol retention, driven by epigenetic modulation of the miR-19a/LDLR axis alongside ABCA1 upregulation and LDLR suppression, collectively impairing proliferation and invasion in gliomas [63].

Notably, cholesterol efflux mechanisms counterbalance uptake to maintain metabolic equilibrium. ABCA1, a cornerstone of HDL biogenesis [64], exemplifies this regulatory duality. In the context of melanoma, M1 macrophage-derived exosomal miR-29c-3p suppresses tumor migration and invasion by orchestrating cholesterol redistribution via the ENPP2-PPARγ-LXRα/β/ABCA1 signaling axis, concurrently modulating extracellular matrix remodeling [65]. Similarly, ABCA8, another ABC transporter critical for sterol efflux and drug resistance, is epigenetically silenced by miR-374b-5p in HCC. This suppression activates the ERK/ZEB1-driven EMT, directly linking cholesterol dysregulation to metastatic progression [66]. GRAM domain-containing protein 1B (GRAMD1B)—a cholesterol transport regulator—emerges as a direct target of miR-4646-5p in triple-negative breast cancer (TNBC). Clinically, elevated miR-4646-5p correlates with improved survival, mechanistically suppressing proliferation and inducing apoptosis through the dual modulation of cholesterol biosynthesis and cytokine signaling networks [21]. These findings delineate miRNA-mediated cholesterol flux regulation as a pleiotropic mechanism influencing tumor–stroma interactions, metabolic reprogramming, and therapeutic vulnerabilities.

#### 3.1.3. MiRNAs Modulate SREBPs in Cancers

SREBPs, master transcriptional regulators of lipid biosynthesis, orchestrate cholesterol, fatty acid, and triglyceride synthesis. The family includes SREBP-1a, SREBP-1c (from the *SREBF-1* gene), and SREBP-2 (from the *SREBF-2* gene) [67,68]. They play key roles in physiological and pathological processes in multiple organs, such as lipid and glucose homeostasis [69]. SREBPs are also critical in diseases like NASH, obesity, and cancer [70,71]. Emerging evidence underscores their dual roles as both effectors and targets of miRNA-mediated regulatory networks in cancer metabolism.

In NSCLC, SREBP-1/SCAP overexpression correlates with aggressive phenotypes, driven by the epigenetic silencing of tumor-suppressive hsa-miR-497-5p. This axis sustains cancer stemness and chemoresistance via SREBP-1/SCAP/FASN lipogenic signaling [72]. Moreover, miR-132 suppresses the expression of SIRT1 and SREBP-1c, leading to a downregulation of their target genes, including *HMGCR* and *FASN*. This action inhibits glioma cell growth, tumorigenicity, invasion, and migration while promoting apoptosis [73]. Notably, SREBP2 serves as the principal transcriptional regulator of cholesterol homeostasis, governing the expression of genes critical for cholesterol biosynthesis, uptake (e.g., LDLR), and storage. In CRC, exosomal miR-1246 secreted by cancer cells targets INSIG1, leading to SREBP2 activation and the subsequent disruption of cholesterol metabolism. The buildup of free cholesterol that follows activates the TLR4/NF-κB/TGF-β pathway. This activation, in turn, spurs hepatic stellate cells (HSCs) into action. Once activated, HSCs contribute to CRC liver metastasis through the TNFSF13/TNFRSF13B axis [74]. According to research by Frank et al., when lactate dehydrogenase B (LDHB) is downregulated in tumor-associated macrophages, it triggers an increase in aerobic glycolysis and lactate production, a response elicited by tumor-derived miR-375. This rise in lactate levels, while inhibiting fatty acid synthesis, unexpectedly activates SREBP2, which in turn stimulates cholesterol biosynthesis within macrophages [75].

Within breast cancer systems, the tumor suppressor miR-874 is downregulated in breast cancer, and its overexpression was found to cause cell cycle arrest, p53 pathway activation, and cell apoptosis by negatively modulating phosphomevalonate kinase (PMVK) and SREBP2 [76]. Furthermore, miR-128 inhibition and miR-223 overexpression directly suppress ABCC5 and UGCG expression, reducing multidrug resistance in breast cancer by enhancing intracellular drug accumulation and cytotoxicity. The altered expression of these miRNAs also decreases lipid raft accumulation, increasing membrane permeability and intracellular tamoxifen retention and thereby reducing tamoxifen resistance. Additionally, miR-128 and miR-223 dysregulation downregulates cholesterol biosynthesis genes, including SREBP1/2, HMGCR, HMGCS1, and LXRα, inhibiting cell survival and proliferation [77].

Prostate cancer studies reveal additional complexity. miR-137 is a potential therapeutic miRNA that, when combined with androgen precursors, can restore the AR-mediated transcription and transactivation axis, maintaining androgenic pathway homeostasis and negatively modulating tumor progression in advanced prostate cancer. Further studies on the miR-137/coregulator (SRC1/SRC2/SRC3)/AR/cholesterol axis are needed to assess its clinical potential [78].

In other words, these findings elucidate a critical regulatory interplay between miRNA networks and cholesterol metabolism in cancer, revealing multilayered mechanisms with translational implications. Notably, the miRNA-mediated modulation of cholesterol biosynthesis (via HMGCR/SQLE), uptake (LDLR), efflux transporters (ABCA1/ABCG8), and master transcriptional regulators of cholesterol metabolism (SREBPs) converges to reprogram tumor growth, stemness, and therapy resistance. Specifically, therapeutic strategies combining cholesterol-lowering agents with miRNA mimics/inhibitors may synergistically disrupt tumor–stroma metabolic symbiosis while overcoming drug tolerance. Future investigations should prioritize isoform-specific miRNA effects, spatial cholesterol distribution in tumor microenvironments, and the biomarker-driven clinical validation of these integrated pathways.

**Table 1 biomedicines-13-01631-t001:** Overview of miRNAs regulating cholesterol metabolism in cancer.

miRNAs	Cancer	Expression	Targets	Pathway	Phenotypes	Ref.
miR-874	breast cancer	down	PMVK, SREBF2	p53	cell cycle arrest apoptosis	[76]
miR-29c-3p	melanoma	/	ENPP2	ENPP2-PPARγ-LXRα/β/ ABCA1	migration invasion ECM remodeling	[65]
miR-146b-5p	bladder cancer	up	FDFT1	/	cisplatin sensitivity	[57]
miR-374b-5p	HCC	up	ABCA8	ERK/ZEB1	Proliferation metastasis	[66]
miR-375	breast cancer	up	LDHB	LDHB/SREBP2	proliferation	[75]
miR-4646-5p	TNBC	/	GRAMD1B	/	proliferation migration	[21]
miR-205	PCa	down	SQLE	SQLE/AR	Proliferation AR inhibitor resistance	[58]
miR-9-5p	breast cancer	up	INSIG1, INSIG2 ATF3	INSIG1/INSIG2/HMGCR ATF3/CH25H	metastasis	[55]
miR-370-3p	cervical cancer	/	DHCR24	/	proliferation migration	[60]
miR-132	glioma	/	SIRT1	SIRT1/SREBP-1c/ HMGCR/FASN	proliferation invasion apoptosis	[73]
miR-1246	CRC	up	INSIG1	INSIG1/SREBP2	proliferation metastasis	[74]
miR-128	breast cancer	/	ABCC5 UGCG	/	drug resistance	[77]
miR-223	breast cancer	/	ABCC5 UGCG	/	drug resistance	[77]
miR-137	prostate cancer	/	SRC-1 SRC-2 SRC-3	SRC1/SRC2/SRC3/AR	proliferation migration invasion	[78]
miR-497-5p	NSCLC	down	SREBP-1 SCAP	SREBP-1/hsa-miR-497/ SCAP/FASN	cell viability cisplatin resistance stemness	[72]
miR-19a	glioma	up	LDLR	PERK/miR-19a/LDLR	proliferation invasion	[63]
miR-363-3p	PAAD	down	SQLE	/	proliferation immune response	[59]

### 3.2. LncRNAs Regulate Cholesterol Metabolism in Cancers

LncRNAs, RNA molecules exceeding 200 nucleotides in length and lacking protein-coding capacity, are widely present in mammals [79,80]. They regulate gene expression at various levels, including epigenetic, transcriptional, post-transcriptional, translational, and post-translational stages, through interactions with mRNAs, DNA, proteins, and miRNAs [79,80]. Importantly, lncRNAs are key regulators of cholesterol metabolism in multiple cancers [20]. For example, Lei et al. performed univariate and multivariate analyses to identify cholesterol-metabolism-related lncRNAs associated with HCC patient prognosis and develop a prognostic signature. They constructed a signature of six such lncRNAs (AC124798.1, AL031985.3, AC103760.1, NRAV, WAC-AS1, and AC022613.1), which can predict HCC prognosis and guide clinical HCC management, including immunotherapy [81]. Herein, we summarize the role and mechanism of lncRNAs in modulating cholesterol metabolism in various cancers (Table 2, Figure 3).

#### 3.2.1. LncRNAs Regulate Cholesterol Metabolism by Modulating Cholesterol Associated with Genes

Emerging evidence delineates lncRNAs as master regulators of cholesterol homeostasis through the direct modulation of cholesterol associated with genes. Notably, 3-Hydroxy-3-methylglutaryl-CoA synthase 2 (HMGCS2)—a mitochondrial rate-limiting enzyme in ketogenesis—plays an oncogenic role in tumors [82]. In CRC, oncogenic lncRNA LOC101928222 is aberrantly overexpressed and correlates with poor prognosis. Mechanistically, LOC101928222 recruits IGF2BP1 to stabilize HMGCS2 transcripts via m6A-dependent mRNA protection, thereby potentiating cholesterol synthesis to drive CRC metastasis and neoangiogenesis [83]. Similarly, pancreatic carcinoma-enriched ZFAS1 promotes tumor aggressiveness by stabilizing HMGCR mRNA through U2AF2-mediated RNA processing. This lncRNA-driven metabolic reprogramming induces lipid accumulation (free fatty acids, cholesterol, triglycerides) and fuels pancreatic carcinoma progression [84]. In addition, in breast cancer stem cells (BCSCs), lnc030 maintains self-renewal capacity and tumorigenicity by cooperating with poly(rC) binding protein 2 (PCBP2) to stabilize SQLE mRNA, increasing cholesterol synthesis. The resulting cholesterol increase activates PI3K/Akt signaling, which regulates BCSC stemness [85]. Linc01711 acts as a scaffold to promote the binding of histone acetyltransferase HBO1 and histone demethylase KDM9. By coordinating the localization of the HBO1/KDM9 complex, Linc01711 specifies the histone modification pattern on target genes, such as lysophosphatidylcholine acyltransferase 1 (LPCAT1), thereby facilitating cholesterol synthesis and contributing to tumor progression [86].

LncRNAs have been shown to be important master transcriptional regulators of cholesterol metabolism. LINC00618 drives HCC EMT and proliferation by stabilizing NSUN2 via ubiquitin–proteasome inhibition. This axis enhances the m5C modification of SREBP2 mRNA, amplifying its stability through YBX1 binding to hyperactivate cholesterol biosynthesis [87]. Retinoid X Receptor Alpha (RXRA), a nuclear receptor, can form heterodimers with receptors, such as LXR and PPAR. These heterodimers bind directly to promoter regions of cholesterol-metabolism-related genes. In hepatoma cells, lncRNA HULC acts as an oncogene. It deregulates lipid metabolism via a miR-9/PPARA/ACSL1 signaling pathway. This pathway is reinforced by a cholesterol/RXRA feed-forward loop, promoting HULC signaling [88].

Beyond enzymatic regulation, lncRNAs modulate therapeutic responses. For instance, linc00339 mediates the antitumor efficacy of cholesterol acyltransferase 1 (ACAT1) inhibitor avasimibe in gliomas, suppressing proliferation and invasiveness through undefined cholesterol-metabolism-dependent mechanisms [89].

#### 3.2.2. LncRNAs Interface with Oncogenic Pathways to Reprogram Cholesterol Metabolism

Emerging evidence delineates lncRNAs as critical modulators of cholesterol metabolism through crosstalk with core oncogenic signaling cascades. Notably, hyperactivated pathways, such as mTORC1, MAPK, and PI3K/AKT, central drivers of tumorigenesis, orchestrate sterol biosynthesis to fuel malignant progression [90,91,92]. In HCC, lncRNA SNHG6 establishes a cholesterol–mTORC1 feed-forward loop by scaffolding the FAF2–mTOR complex at ER–lysosome contact sites. This molecular interplay potentiates cholesterol-dependent mTORC1 lysosomal recruitment and activation, thereby coupling sterol synthesis to the NAFLD–HCC transition by self-amplifying metabolic reprogramming [90]. Similarly, LINC01605 drives pancreatic ductal adenocarcinoma (PDAC) progression by hijacking LIN28B-mediated cholesterol flux to hyperactivate mTOR signaling, underscoring the pathway’s central role in lipid-driven oncogenesis [93].

The MYC oncogene, a master transcriptional regulator of proliferation and metabolism [94], converges with lncRNAs to amplify sterol biosynthesis. In MYC-driven lymphomas, KTN1-AS1—directly transactivated by MYC—co-regulates MYC-target genes while co-activating cholesterol synthesis effectors, such as SQLE. The depletion of KTN1-AS1 attenuates B-cell lymphoma proliferation, revealing its indispensable role in MYC-mediated metabolic adaptation [95].

Expanding this paradigm, the Hippo–YAP pathway intersects with cholesterol metabolism [96]. The pharmacological inhibition of HMGCR with Simvastatin suppresses SNHG29, a lncRNA stabilizing YAP, by impeding its ubiquitin–proteasomal degradation. This axis downregulates PD-L1 transcription, thereby enhancing antitumor immunity while disrupting the SNHG29–YAP–PD–L1 circuit that sustains immunosuppressive microenvironments [97].

Kruppel-like factor 4 (KLF4), a zinc finger transcription factor and member of the Spl/Kruppel-like zinc finger family, is pivotal in regulating cell proliferation, differentiation, and apoptosis, and it is implicated in various diseases, including cancer [98]. In CRC, LINC01915 antagonizes tumor–stroma crosstalk by competitively binding miR-92a-3p to upregulate KLF4. This transcription factor elevates CH25H expression, suppressing extracellular vesicle uptake by normal fibroblasts and inhibiting CAF-driven angiogenesis—a dual mechanism decoupling cholesterol metabolism from stromal activation [99].

Intriguingly, estrogen-regulated LincNORS fine-tunes sterol/steroid biosynthesis through RALY-dependent RNA–protein interactions. This lncRNA globally represses cholesterol pathway components, positioning RALY as a rheostat integrating hormonal signaling with lipid homeostasis [100].

In conclusion, these findings highlight that lncRNAs play a significant role in regulating cholesterol metabolism, both by modulating genes associated with cholesterol and by interacting with oncogenic pathways. This dual function of lncRNAs not only advances our understanding of cancer metabolism but also paves the way for developing innovative treatment strategies.

**Table 2 biomedicines-13-01631-t002:** Overview of lncRNAs regulating cholesterol metabolism in cancer.

lncRNAs	Cancer	Expression	Regulatory Mechanism	Phenotypes	Ref.
LOC101928222	CRC	up	collaborates with IGF2BP1 to stabilize HMGCS2 mRNA	migration, invasion, angiogenesis	[83]
HULC	HCC	up	ceRNA for miR-9 increases PPARA expression	proliferation	[88]
LINC00618	HCC	up	Interacts with NSUN2 and regulates SREBP2 mRNA stability	proliferation, migration EMT	[87]
SNHG6	HCC	up	Interacts with FAF2–mTOR complex	proliferation progression	[90]
linc00339	glioma	/	/	proliferation, migration invasion	[89]
SNHG29	CRC	up	Interacts with YAP and regulates PD-L1 expression	tumor immunity	[97]
lnc030	breast cancer	up	cooperates with PCBP2 to stabilize SQLE mRNA	stemness	[85]
ZFAS1	pancreatic carcinoma	up	interacts with U2AF2 to stabilize HMGCR mRNA	proliferation, invasion	[84]
KTN1-AS1	Burkitt lymphoma	up	co-regulates Myc-target genes	proliferation	[95]
lincNORS	breast cancer	/	interacts with RALY	/	[100]
Linc01711	gastric cancer	up	coordinates the localization of the HBO1/KDM9 complex	proliferation invasion metastasis	[86]
LINC01915	CRC	down	ceRNA for miR-92a-3p to regulate the KLF4/CH25H axis	conversion of NF into CAF	[99]
LINC01605	PDAC	up	interacts with LIN28B and activates the mTOR axis	proliferation migration	[93]

### 3.3. CircRNAs Orchestrate Cholesterol Metabolic Reprogramming in Cancers

CircRNA, a closed-loop RNA molecule formed by backsplicing, exhibits remarkable stability and functional versatility in eukaryotic systems [100,101]. Emerging evidence delineates their multifaceted roles as miRNA sponges, transcriptional regulators, and scaffolds for protein complexes, with growing implications for oncogenic cholesterol metabolism [100,101,102]. In the current study, the roles and mechanisms of lncRNAs in modulating cholesterol metabolism in various cancers are summarized in Table 3.

In PAAD, circ_0124346 emerges as a metabolic driver aberrantly overexpressed in tumors and cell lines. Mechanistically, this circRNA hijacks the miR-223-3p/ACSL3 axis to amplify lipid synthesis—ACSL3 catalyzes fatty acyl-CoA esterification, fueling triglyceride and cholesterol biogenesis—while correlating with aggressive tumor dimensions [103]. Similarly, circLDLR is upregulated in CRC and linked to poor prognosis. Functional analyses reveal its role in stabilizing SOAT1 via miR-30a-3p sequestration, thereby augmenting cholesterol esterification to propel CRC growth and metastasis [104]. In prostate cancer, circFAM126A is aberrantly overexpressed and associated with a poor prognosis. CircFAM126A exerts its effects by targeting miR-505-3p, which in turn regulates calnexin (CANX). Upregulating miR-505-3p or inhibiting CANX suppresses both cholesterol synthesis and the malignant progression of prostate cancer cells [105].

The circRNA–cholesterol axis extends to gastric malignancies, where circ_0000182 promotes sterol overproduction in stomach adenocarcinoma (STAD) by sponging miR-579-3p to derepress SQLE, a rate-limiting enzyme in cholesterol biosynthesis. This regulatory loop correlates with enhanced tumor proliferation and clinical progression [106]. Intriguingly, circRIC8B rewires lipid metabolism in CLL by antagonizing the miR-199b-5p-mediated degradation of LPL mRNA. This circuit enhances triglyceride hydrolysis and cholesterol uptake, synergizing with ezetimibe’s lipid-lowering effects to sustain CLL survival [19].

In TNBC, circMyc (hsa_circ_0085533) drives lipid accumulation by stabilizing SREBP1 transcripts via HuR protein recruitment. Notably, circMyc overexpression elevates triglycerides, cholesterol, and lipid droplets, establishing it as a master regulator of TNBC metabolic plasticity [107]. Hypoxia-inducible circRNAs further expand this paradigm; circINSIG1 encodes a functional 121-aa peptide (circINSIG1-121) that recruits the CUL5-ASB6 E3 ligase to degrade INSIG1 via K48-linked ubiquitination. This cholesterol–biosynthetic switch accelerates CRC proliferation and metastatic dissemination [108].

**Table 3 biomedicines-13-01631-t003:** Overview of circRNAs regulating cholesterol metabolism in cancer.

circRNAs	Cancer	Expression	Regulatory Mechanism	Phenotypes	Ref.
circFAM126A	PCa	up	ceRNA for miR-505-3p and increases CANX expression	proliferation, apoptosis invasion migration	[105]
Circ_0000182	STAD	up	ceRNA for miR-579-3p and increases SQLE expression	proliferation	[106]
circ_0124346	PAAD	up	ceRNA for miR-223-3p and increases ACSL3 expression	proliferation	[103]
circLDLR	CRC	up	ceRNA for miR-30a-3p and increases SOAT1 expression	proliferation invasion migration	[104]
circMyc	TNBC	up	interacts with HuR to stabilize SREBP1 mRNA	proliferation invasion	[107]
circRIC8B	CLL	up	ceRNA for miR-199b-5p and increases LPL expression	proliferation	[19]
circINSIG1	CRC	up	promotes the degradation of INSIG1	proliferation metastasis	[108]

Collectively, these findings position circRNAs as central mechanistic conduits bridging cholesterol metabolic reprogramming with oncogenic progression while simultaneously exposing actionable therapeutic nodes for RNA-based intervention strategies targeting lipid-driven malignancies. This dual functionality not only redefines circRNAs as master regulators of sterol–oncogenic crosstalk but also pioneers novel therapeutic paradigms by leveraging RNA interference (RNAi) or CRISPR-based circRNA modulation to disrupt tumor metabolic addiction.

### 3.4. piRNAs and tRNAs Modulate Cholesterol Metabolism in Cancers

PiRNAs are small non-coding RNAs that specifically bind PIWI proteins [109,110]. They are 23–31 nt long, with a 5′-terminal uridine (U) and a 2′-O-methylated 3′-end. PiRNAs are transcribed from long single-stranded templates and are Dicer-independent. They are 24–32 nt in length, with a 5′-terminal U and, often, an adenine (A) at position 10, and their genomic sequences cluster [109,110]. Importantly, piRNAs have emerged as key players in cancer biology [111,112]. They influence tumor metabolism through diverse mechanisms, such as gene regulation, epigenetic control, and metabolic pathway interactions [112]. For instance, piR-33422 is downregulated in tongue squamous cell carcinoma and has a binding site within the 5′-UTR of FDFT1, a critical enzyme in cholesterol synthesis [113]. Similarly, elevated piRNA-137463 levels are associated with unfavorable prognoses in lung adenocarcinoma (LUAD) patients. Inhibiting piRNA-137463 not only reduces LUAD cell proliferation, migration, and invasion but also enhances T cell cytotoxicity through increased IFN-γ secretion. It further disrupts cholesterol metabolism, reducing intracellular cholesterol, lipid raft content, and PD-L1 expression in LUAD cells. Moreover, piRNA-137463 inhibition increases the stability and expression of LOC100128494, thereby modulating INSIG1 levels via a ceRNA network involving LOC100128494 and miR-24-3p. Notably, the effects of piRNA-137463 in LUAD cells are contingent on the expression of LOC100128494 and INSIG1. Finally, treatment with AntagomiR-137463 to suppress piRNA-137463 expression inhibits tumor growth and metastasis in nude mice via LOC100128494 and enhances the response of LUAD to anti-PD-1 therapy in immune-competent mice [114].

Transfer RNA (tRNA), a small non-coding RNA, primarily functions in protein synthesis by transporting amino acids to ribosomes and pairing with mRNA codons. It has a cloverleaf structure, it is 70–90 nucleotides long, it carries specific amino acids at its 3′-end, and it has an anticodon at its 5′-end that complements mRNA codons. tRNA also contains various modified nucleotides essential for stability and function. Different tRNAs transport specific amino acids, with their anticodons specifically recognizing different mRNA codons [115,116]. tRNA plays a key role in cholesterol metabolism by accelerating protein synthesis to support rapid tumor cell proliferation [116,117]. Wang et al. revealed that TRMT6 and TRMT61A, components of the m1A methyltransferase complex, increase m1A methylation in specific tRNAs. This enhances PPARδ translation, stimulating cholesterol synthesis and activating Hedgehog signaling, thereby promoting the self-renewal of liver cancer stem cells and tumorigenesis [118]. Additionally, TRMT61A, an m1A “writer” gene, boosts the tumor-killing ability of CD8+ T cells by regulating cholesterol biosynthesis. Trmt61a deletion in CD8+ T cells impairs their tumor-killing function in both in vivo and in vitro settings. Mechanistically, tRNA m1A modification enhances antitumor immunity in CD8+ T cells by upregulating the translation of ATP citrate lyase, a key enzyme in cholesterol biosynthesis [119].

In conclusion, the participation of piRNAs and tRNAs in regulating cholesterol metabolism, either directly or indirectly, provides a potential pathway for enhancing our comprehension of tumor metabolism and innovating therapeutic strategies.

## 4. Conclusions and Perspectives

Expanding evidence underscores cholesterol metabolism’s pivotal role in tumor biological processes, such as growth, metastasis, drug resistance, and immune evasion, which await further clarification across various cancer types [6,40,41]. The regulation of cholesterol metabolism in tumors is intricate, involving multiple intracellular and extracellular factors, including key signaling molecules, ncRNAs, and the tumor microenvironment [9,20]. This review centers on ncRNA dysregulation and its impact on cholesterol metabolism in cancers. Specifically, miRNAs, lncRNAs, circRNAs, piRNAs, and tRNAs can directly or indirectly modulate cholesterol metabolism in cancers.

Despite notable advancements in the field, several fundamental questions regarding the role of ncRNAs in tumor cholesterol metabolism remain unanswered. While numerous studies have elucidated the expression patterns of cholesterol-metabolism-associated ncRNAs across various cancer types [120,121], their high specificity and accuracy suggest that they could serve as promising biomarkers for cancer diagnosis, a potential that warrants further investigation [120,121]. This review examines the roles of miRNAs, lncRNAs, circRNAs, piRNAs, and tRNAs in regulating cholesterol metabolism. However, the impact of other ncRNAs, such as snRNAs and snoRNAs, on cholesterol metabolism in cancers remains largely unexplored and necessitates additional research.

Developing antitumor strategies targeting the ncRNA–cholesterol metabolism axis is of critical importance. The specific molecular mechanisms through which ncRNAs regulate cholesterol metabolism are still not fully understood. Many studies only describe the expression correlation between ncRNAs and cholesterol-metabolism-related genes, lacking in-depth analysis of downstream signaling and metabolic pathways. Additionally, research on the ncRNA regulation of cholesterol metabolism spans multiple disciplines, including molecular biology, metabolomics, and oncology, yet interdisciplinary, collaborative research is currently limited. This disciplinary divide restricts a comprehensive understanding of ncRNA’s regulatory mechanisms. Finally, although the role of ncRNAs in tumor cholesterol metabolism has been widely reported, translating these findings into clinical applications remains challenging. For example, the specificity and effectiveness of ncRNAs as therapeutic targets still require further validation.

To advance this field, a multi-omics approach integrating data from transcriptomics, proteomics, and metabolomics can provide a comprehensive analysis of the ncRNA regulatory network in cholesterol metabolism. This approach can reveal the complex interactions between ncRNAs and other biomolecules, offering a more holistic view of their role in tumors. Moreover, studying the dynamic changes and regulatory mechanisms of ncRNAs under different physiological and pathological conditions, such as hypoxia and energy stress, is crucial for understanding their impact on cholesterol metabolism and tumor development. Clinical translation research should also be strengthened with large-scale clinical sample analyses to verify the diagnostic value of ncRNAs and explore targeted therapeutic strategies based on ncRNAs, including assessments of their safety and effectiveness in clinical treatment.

In summary, the regulatory role of ncRNAs in tumor cholesterol metabolism holds significant research and clinical potential. Strengthening interdisciplinary collaboration, conducting in-depth mechanistic research, and promoting clinical translation are essential steps to fill current research gaps and achieve the widespread application of ncRNAs in tumor treatment. Future research must be dedicated to addressing these gaps to fully harness the potential of ncRNAs in cancer therapy.

## Figures and Tables

**Figure 1 biomedicines-13-01631-f001:**
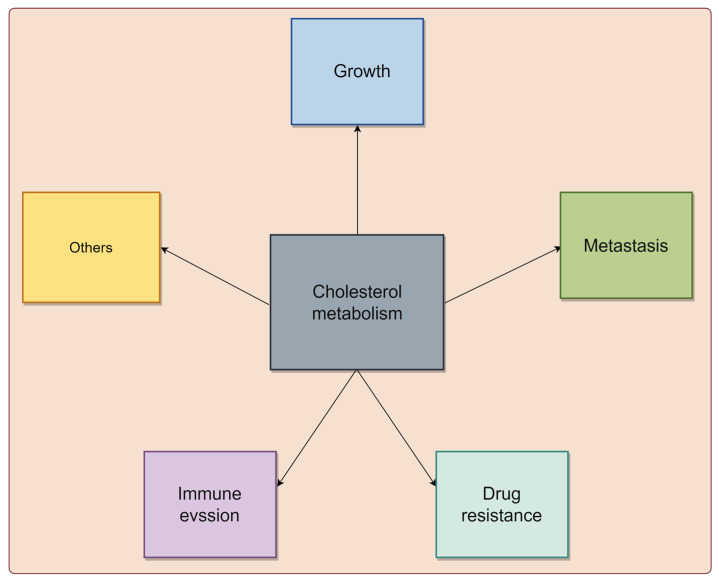
The biological functions of cholesterol metabolism in cancer. Cholesterol metabolism plays crucial roles in modulating tumor growth, metastasis, drug resistance, and immune evasion, highlighting their impact on tumor progression. The graphic was created using Figdraw (www.figdraw.com).

**Figure 2 biomedicines-13-01631-f002:**
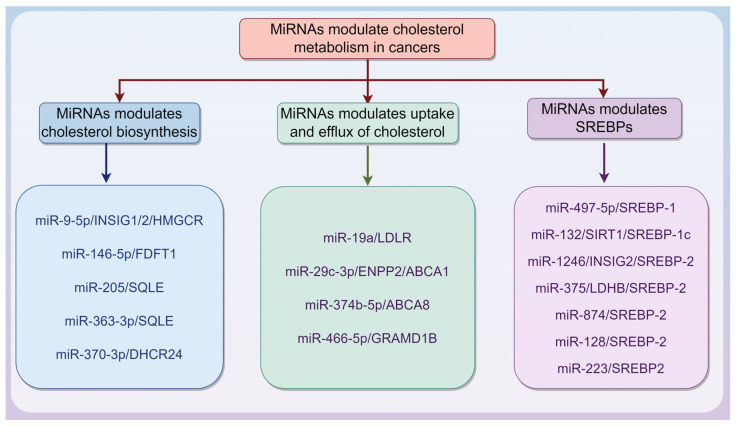
microRNA (miRNA) regulates cholesterol metabolism in cancers via different mechanisms. This is described in Table 1. The graphic was created using Figdraw (www.figdraw.com).

**Figure 3 biomedicines-13-01631-f003:**
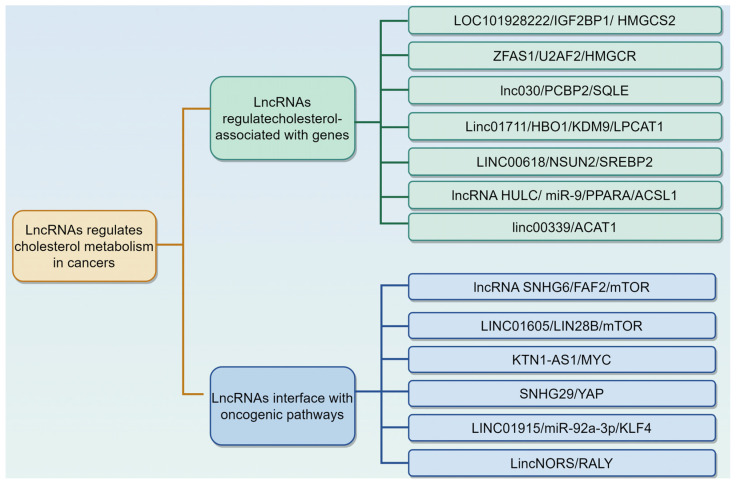
Long non-coding RNAs (lncRNAs) regulate cholesterol metabolism in cancers via different mechanisms. This is described in Table 2. The graphic was created using Figdraw (www.figdraw.com).

## Data Availability

Not applicable.

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
