# Peer review of "Non-Coding RNAs as Critical Modulators of Cholesterol Metabolism in Cancer"

_biomedicines, 2025, doi:10.3390/biomedicines13071631_

Round 1

Reviewer 1 Report

Comments and Suggestions for Authors

The manuscript is well written and contain valuable information regarding the role of non-coding RNA in modulation of cholesterol metabolism in caner. The manuscript offers a comprehensive analysis of cholesterol metabolism reprogramming in tumor progression, influencing growth, metastasis, drug resistance, and immune evasion. The authors use update references.

Minor points:

1-in section 3.2.1 line 4-6 are not scientifically correct.

“Notably, 3-Hydroxy-3-methylglutaryl-CoA synthase 2 (HMGCS2)—a mitochondrial rate-limiting enzyme in ketogenesis and cholesterol biosynthesis—serves dual roles in energy adaptation 334 during nutrient deprivation and sterol homeostasis maintenance”

HMGCS2,a mitochondrial rate-limiting enzyme in ketogenesis,—plays e key role in energy adaptation during nutrient deprivation but is not involved in cholesterol biosynthesis.

-Cholesterol biosynthesis take place in the cytosol and the enzyme involved there is HMGCS1, not HMGCS2.

-Dual roles (as the authors suggested) do exist but only within energy metabolism (e.g. ketone production during prolong fasting), not in sterol homeostasis.

-The enzyme responsible for sterol homeostasis is HMG-CoA reductase, not HMGCs2.

2- It is recommended that the authors also use the following fully relevant reference:

The role of noncoding RNAs in cancer lipid metabolism. Front Oncol. 2022 Nov 14;12:1026257. doi: 10.3389/fonc.2022.1026257

Author Response

The manuscript is well written and contain valuable information regarding the role of non-coding RNA in modulation of cholesterol metabolism in caner. The manuscript offers a comprehensive analysis of cholesterol metabolism reprogramming in tumor progression, influencing growth, metastasis, drug resistance, and immune evasion. The authors use update references.

Minor points:

Comments 1: [1-in section 3.2.1 line 4-6 are not scientifically correct.

“Notably, 3-Hydroxy-3-methylglutaryl-CoA synthase 2 (HMGCS2)—a mitochondrial rate-limiting enzyme in ketogenesis and cholesterol biosynthesis—serves dual roles in energy adaptation 334 during nutrient deprivation and sterol homeostasis maintenance”

HMGCS2,a mitochondrial rate-limiting enzyme in ketogenesis,—plays e key role in energy adaptation during nutrient deprivation but is not involved in cholesterol biosynthesis.

-Cholesterol biosynthesis take place in the cytosol and the enzyme involved there is HMGCS1, not HMGCS2.

-Dual roles (as the authors suggested) do exist but only within energy metabolism (e.g. ketone production during prolong fasting), not in sterol homeostasis.

-The enzyme responsible for sterol homeostasis is HMG-CoA reductase, not HMGCs2.]

Response 1: [We are grateful for the reviewer’s insightful suggestion. In response, we have thoroughly revised the introduction of 3-Hydroxy-3-methylglutaryl-CoA synthase 2 (HMGCS2). We described as: Notably, 3-Hydroxy-3-methylglutaryl-CoA synthase 2 (HMGCS2)—a mitochondrial rate-limiting enzyme in ketogenesis—plays oncogenic role in tumors (Life Sci . 2023 Sep 1:328:121827).]

Comments 2:[2- It is recommended that the authors also use the following fully relevant reference:

The role of noncoding RNAs in cancer lipid metabolism. Front Oncol. 2022 Nov 14;12:1026257. doi: 10.3389/fonc.2022.1026257]

 Response 2: [Thanks. We have used the reference.]

Reviewer 2 Report

Comments and Suggestions for Authors

With this manuscript, Zhang et al. highlight recent advancements in understanding the regulatory mechanisms of non-coding RNAs in cancer-associated cholesterol metabolism, emphasizing their potential as therapeutic targets. However, the current version of the manuscript does not meet the standards required for publication in this journal. Substantial revisions are necessary before it can be reconsidered for acceptance.

Major

Insufficient Original Synthesis and Novelty: The manuscript lacks unique synthesis and new hypotheses, yet it is a well-documented summary. It mostly summarizes current research without putting out any novel models or mechanistic insights. Repetition and Redundancy: With little integration across ncRNA types, several sections—particularly those that describe miRNAs, lncRNAs, and circRNAs—repeat similar content (e.g., detailing their involvement in cholesterol metabolism and cancer). Cohesion is hampered, and needless length is added. Over-reliance on Tables Without Critical Analysis: Although the tables provide useful information, the manuscript depends too much on them to provide important details. The table material is frequently repeated throughout the text without further explanation or integration. Inadequate Organization and Section Transition: There is a mechanical flow between themes (e.g., miRNAs → lncRNAs → circRNAs). The review feels disjointed because transitions lack thematic progression and narrative connection. Insufficient Critical Evaluation of Research: The article provides a summary of the literature's findings without assessing their merits, shortcomings, or inconsistencies. Conflicting evidence, methodological issues, or the validity of the results should all be discussed in a critical review. Inadequate Conclusion and Viewpoint: Most of the issues made earlier are restated in the conclusion section. It makes no relevant or thorough recommendations for specific future directions or research gaps.

Minor

Figure 1 Is Underexplained and Generic. Although the graphic is conceptually helpful, the legend is vague and uninformative. It ought to be incorporated into the text more thoroughly with a discussion that is cited. Reference formatting that is inconsistent. The text occasionally lacks context and references are positioned inconsistently. Make sure every citation backs up a particular assertion rather than providing background information. Insufficient Attention to Clinical Significance. Although therapeutic implications are mentioned in the review, it doesn't go far enough in discussing how results might be converted into prognoses, therapies, or diagnostics.

Comments on the Quality of English Language

Quality of English Language is not satisfactory

Author Response

With this manuscript, Zhang et al. highlight recent advancements in understanding the regulatory mechanisms of non-coding RNAs in cancer-associated cholesterol metabolism, emphasizing their potential as therapeutic targets. However, the current version of the manuscript does not meet the standards required for publication in this journal. Substantial revisions are necessary before it can be reconsidered for acceptance.

Major

Comments 1: [Insufficient Original Synthesis and Novelty: The manuscript lacks unique synthesis and new hypotheses, yet it is a well-documented summary. It mostly summarizes current research without putting out any novel models or mechanistic insights.]

Response 1: [Thank you for your insightful query. In the introduction, we discussed the pivotal role of cholesterol metabolism in cancer, highlighting its significance in driving the search for novel molecules and strategies targeting cholesterol pathways. This focus has led to substantial advancements in recent years. Therefore, a deeper understanding of the regulatory mechanisms governing cholesterol metabolism is essential for developing innovative therapeutic strategies aimed at modulating cholesterol levels and mitigating cancer risks. Given the critical roles of non-coding RNAs (ncRNAs) in various biological functions, including cholesterol metabolism, it is surprising that few systematic reviews have addressed their functions in this context. This review aims to fill this gap by summarizing the current understanding of how ncRNAs regulate cholesterol metabolism in cancer. We anticipate that these insights will facilitate the identification of novel therapeutic strategies for cancer treatment.

In the conclusion, we have provided the summary to more clearly articulate the unique contributions of our manuscript. This review centers on the dysregulation of ncRNAs and its impact on cholesterol metabolism in cancers. Specifically, miRNAs, lncRNAs, circRNAs, piRNAs, and tRNAs can directly or indirectly modulate cholesterol metabolism in cancer. Developing anti-tumor strategies targeting the ncRNA-cholesterol metabolism axis is of critical importance. The specific molecular mechanisms by which ncRNAs regulate cholesterol metabolism are still not fully understood. Many studies only describe the expression correlation between ncRNAs and cholesterol metabolism-related genes, lacking in-depth analysis of downstream signaling and metabolic pathways. Additionally, research on ncRNA regulation of cholesterol metabolism spans multiple disciplines, including molecular biology, metabolomics, and oncology, yet interdisciplinary collaborative research is currently limited. This disciplinary divide restricts a comprehensive understanding of ncRNA regulatory mechanisms. Finally, although the role of ncRNAs in tumor cholesterol metabolism has been widely reported, translating these findings into clinical applications remains challenging. For example, the specificity and effectiveness of ncRNAs as therapeutic targets still require further validation.]

Comments 2:[Repetition and Redundancy: With little integration across ncRNA types, several sections—particularly those that describe miRNAs, lncRNAs, and circRNAs—repeat similar content (e.g., detailing their involvement in cholesterol metabolism and cancer). Cohesion is hampered, and needless length is added.]

Response 2: [This is a nice suggestion. We have streamlined the content across sections, particularly those discussing miRNAs, lncRNAs, and circRNAs, to eliminate redundancy and enhance coherence. We have revised the relevant section as follows: ncRNAs, including miRNA, lncRNA, and circRNA, have been extensively documented as key regulators of gene expression, cellular physiology, and tumor development. Notably, these ncRNAs also play a pivotal role in modulating cholesterol metabolism within cancer cells (Front Oncol . 2022 Nov 14:12:1026257.).]

Comments 3: [Over-reliance on Tables Without Critical Analysis: Although the tables provide useful information, the manuscript depends too much on them to provide important details. The table material is frequently repeated throughout the text without further explanation or integration.]

Response 3: [Thank you for your insightful feedback. We have carefully reviewed the manuscript to identify instances where table material was unnecessarily repeated in the text. We have discussed the significance of the findings, highlighting key trends, and explaining how these insights contribute to the broader context of our research.]

Comments 4: [Inadequate Organization and Section Transition: There is a mechanical flow between themes (e.g., miRNAs → lncRNAs → circRNAs). The review feels disjointed because transitions lack thematic progression and narrative connection.]

Response 4: [Thank you for your insightful feedback. We have carefully organized the manuscript to cover the major classes of ncRNAs involved in cholesterol metabolism and cancer, namely miRNAs, lncRNAs, circRNAs, piRNAs and tRNAs. This sequential arrangement was chosen to provide a clear and comprehensive overview of each ncRNA type, allowing readers to understand their distinct roles and mechanisms in the context of cholesterol metabolism and cancer progression.]

Comments 5: [Insufficient Critical Evaluation of Research: The article provides a summary of the literature's findings without assessing their merits, shortcomings, or inconsistencies. Conflicting evidence, methodological issues, or the validity of the results should all be discussed in a critical review.]

Response 5: [We thank the reviewer for the helpful suggestion. We have discussed the roles and mechanisms of ncRNAs in regulating cholesterol metabolism in cancers. Moreover, we have discussed the insufficient critical evaluation of research. For example, developing anti-tumor strategies targeting the ncRNA-cholesterol metabolism axis is of critical importance. The specific molecular mechanisms by which ncRNAs regulate cholesterol metabolism are still not fully understood. Many studies only describe the expression correlation between ncRNAs and cholesterol metabolism-related genes, lacking in-depth analysis of downstream signaling and metabolic pathways. Additionally, research on ncRNA regulation of cholesterol metabolism spans multiple disciplines, including molecular biology, metabolomics, and oncology, yet interdisciplinary collaborative research is currently limited.]

Comments 6: [Inadequate Conclusion and Viewpoint: Most of the issues made earlier are restated in the conclusion section. It makes no relevant or thorough recommendations for specific future directions or research gaps.

Response 6: [This is a nice suggestion. We have provided the recommendations for specific future directions or research gaps.]

Minor

Comments 7: [Figure 1 Is Underexplained and Generic. Although the graphic is conceptually helpful, the legend is vague and uninformative. It ought to be incorporated into the text more thoroughly with a discussion that is cited.]

Response 7: [Thanks. We agreed that a more detailed explanation and better incorporation into the discussion would significantly improve the figure's utility and the overall coherence of the manuscript.We have enhanced the clarity and integration of this figure into the text.]

Comments 8:[Reference formatting that is inconsistent. The text occasionally lacks context and references are positioned inconsistently. Make sure every citation backs up a particular assertion rather than providing background information.]

Response 8:[Thanks. We have revised the reference formatting.]

Comments 9:[Insufficient Attention to Clinical Significance. Although therapeutic implications are mentioned in the review, it doesn't go far enough in discussing how results might be converted into prognoses, therapies, or diagnostics.]

Response 9: [Thanks. We have discussed the clinical significance. We discussed as: Clinical translation research should also be strengthened, with large-scale clinical sample analyses to verify the diagnostic value of ncRNAs and the exploration of targeted therapeutic strategies based on ncRNAs, including assessments of their safety and effectiveness in clinical treatment.]

Reviewer 3 Report

Comments and Suggestions for Authors

Nowadays non-coding RNAs are very important topic in the RNA field. The authors’ choice to perform a study investigating Non-Coding RNAs as Critical Modulators of Cholesterol Metabolism in Cancer is novel and adds significant information to the literature. My opinion the manuscript is not suited for publication in its present form and the authors should perform some revisions regarding specific points as highlighted herein.

I think you should make revision of the manuscript by a native English speaker in order to improve its performance with regard to syntax, grammar and phraseology.

Abstract section:

This abstract section is not well-written and adequately presented. Authors should clearly present the aim of the paper and the knowledge gap driving conduction of this study.

Authors should make revision to all figures; for example, figure 1 does not effectively convey the intended expression. The authors have expressed the effects of non-coding RNAs. They can provide figures to have more effective explanation.

According to author guidelines, abbreviations should be given in brackets after their first mention in the text, and used thereafter. Please extensively revise the manuscript in the main text.

The Conclusions and perspectives is poorly presented and the authors' efforts to summarize the evidence in this area of interest are inadequate. This should be revised.

I hope the suggestions for the better of this prepared manuscript will be useful to you.

Comments on the Quality of English Language

I think you should make revision of the manuscript by a native English speaker in order to improve its performance with regard to syntax, grammar and phraseology.

Author Response

Nowadays non-coding RNAs are very important topic in the RNA field. The authors’ choice to perform a study investigating Non-Coding RNAs as Critical Modulators of Cholesterol Metabolism in Cancer is novel and adds significant information to the literature. My opinion the manuscript is not suited for publication in its present form and the authors should perform some revisions regarding specific points as highlighted herein.

Comments 1: [I think you should make revision of the manuscript by a native English speaker in order to improve its performance with regard to syntax, grammar and phraseology.]

Response 1: [The manuscript was polished for English language, grammar, punctuation, spelling, and overall style by MDPI Author Services.]

Comments 2: [Abstract section:

This abstract section is not well-written and adequately presented. Authors should clearly present the aim of the paper and the knowledge gap driving conduction of this study.]

Response 2: [This is a nice suggestion. We have revised the abstract section to clearly articulate the aim of the paper and the knowledge gap that motivated this study.]

Comments 3: [Authors should make revision to all figures; for example, figure 1 does not effectively convey the intended expression. The authors have expressed the effects of non-coding RNAs. They can provide figures to have more effective explanation.]

Response 3: [Thank you for your valuable suggestion. We have enhanced the clarity and integration of Figure 1 into the text. Moreover, we have added Figure 2 and Figure 3, which illustrate the effects of miRNAs and lncRNAs on cholesterol metabolism, respectively.]

Comments 4: [According to author guidelines, abbreviations should be given in brackets after their first mention in the text, and used thereafter. Please extensively revise the manuscript in the main text.]

Response 4: [Thanks. We have revised the abbreviations.]

Comments 5: [The Conclusions and perspectives is poorly presented and the authors' efforts to summarize the evidence in this area of interest are inadequate. This should be revised.]

Response 5: [This is a nice suggestion. We have revised the conclusions and perspectives and provided the recommendations for specific future directions or research gaps.]

Comments 6: [Comments on the Quality of English Language

I think you should make revision of the manuscript by a native English speaker in order to improve its performance with regard to syntax, grammar and phraseology.]

Response 6: [The manuscript was polished for English language, grammar, punctuation, spelling, and overall style by MDPI Author Services.]